# Fabrication of Porous Carbon Nanofibers from Polymer Blends Using Template Method for Electrode-Active Materials in Supercapacitor

**DOI:** 10.3390/molecules28052228

**Published:** 2023-02-27

**Authors:** He Wang, Lan Yao, Hongmei Zuo, Fangtao Ruan, Hongjie Wang

**Affiliations:** 1School of Textile and Garment, Anhui Polytechnic University, Wuhu 241000, China; 2China National Textile and Apparel Council Key Laboratory of Flexible Devices for Intelligent Textile and Apparel, Soochow University, Suzhou 215123, China; 3Advanced Fiber Materials Engineering Research Center of Anhui Province, Anhui Polytechnic University, Wuhu 241000, China

**Keywords:** porous carbon nanofiber, polymer, blend, electrode material, supercapacitor

## Abstract

Porous carbon nanofibers (PCNFs) with excellent physical and chemical properties have been considered candidate materials for electrodes used in supercapacitors. Herein, we report a facile procedure to fabricate PCNFs through electrospinning blended polymers into nanofibers followed by pre-oxidation and carbonization. Polysulfone (PSF), high amylose starch (HAS), and phenolic resin (PR) are used as three different kinds of template pore-forming agents. The effects of pore-forming agents on the structure and properties of PCNFs have been systematically studied. The surface morphology, chemical components, graphitized crystallization, and pore characteristics of PCNFs are analyzed by scanning electron microscopy (SEM), X-ray photoelectron spectroscopy (XPS), X-ray diffraction (XRD), and nitrogen adsorption and desorption test, respectively. The pore-forming mechanism of PCNFs is analyzed by differential scanning calorimetry (DSC) and thermogravimetric analysis (TGA). Fabricated PCNF-R have a specific surface area as high as ~994 m^2^/g, a total pore volume as high as ~0.75 cm^3^/g, and a good graphitization degree. When PCNF-R are used as active materials to fabricate into electrodes, the PCNF-R electrodes show a high specific capacitance ~350 F/g, a good rate capability ~72.6%, a low internal resistance ~0.55 Ω, and an excellent cycling stability ~100% after 10,000 charging and discharging cycles. The design of low-cost PCNFs is expected to be widely applicable for the development of high-performance electrodes for an energy storage field.

## 1. Introduction

Supercapacitors with fast charging and discharging ability, high power density, and long-life span have been regarded as green energy storage devices, which play important roles in the life of human beings [1,2,3,4,5]. The current collector, active material, electrolyte, and separator are the main components of supercapacitors [6]. The supercapacitor electrode can be fabricated by dipping, spraying, plating, and depositing active materials, e.g., carbon-based materials, conductive polymer-based materials, and metal oxide-based materials on the surface of the current collector [7,8,9,10,11,12,13]. Moreover, the electrode-active material determines the electrochemical performance of supercapacitors. Therefore, the exploitation of electrode-active material has become meaningful research at present.

Electrospinning is a relatively general technology for fabricating ultra-fine one-dimensional nanofibers, two-dimensional nonwoven films, and three-dimensional scaffolds [14,15,16,17]. Carbon nanofibers (CNFs) can be fabricated using electrospun nanofibers (NFs) followed by high-temperature sintering [18]. To enhance the performance of CNFs and expand their application fields, porous carbon nanofibers (PCNFs) with excellent physical–chemical properties have been exploited. The PCNFs have high specific area and porosity, outstanding electrical conductivity, good stability, and a simple manufacturing process, which is considered to be an ideal candidate for making high-performance supercapacitors [19,20,21,22].

Currently, there are many methods to fabricate PCNFs, such as the physical method, chemical method, and template method [23,24,25,26,27]. However, the fabrication process of physical and chemical methods is more complex, and the energy consumption is very high, resulting in high cost, which is not conducive to large-scale manufacturing. In early research, although high specific surface areas can be obtained by physical methods (CO_2_, steam as activator), the specific capacitance of the electrode is lower than 300 F/g due to the tedious microporous structure [23]. To enhance the specific capacitance of PCNFs-based electrode, a chemical method was developed. PCNFs prepared by chemical activators such as KOH, H_3_PO_4_, and ZnCl not only have high specific surface areas and rich micro-meso pore structures but also have active elements doped in carbon nanofibers, effectively improving specific capacitance [24,25,26]. However, the excessive use of chemical activators is not conducive to environmental protection, and the strong corrosivity will damage the equipment. These adverse factors limit the development of chemical activators. By comparison, the template method is a more convenient and commonly used method to fabricate PCNFs. PCNFs can be fabricated by carbonizing blended NFs containing a polymer-derived carbon source and pore-forming agent. The polymer precursor is converted into carbon molecules after high-temperature carbonization, and the pore-forming agent is in situ degraded, leaving a large number of pores. However, there are a variety of polymer pore-forming agents available. Some thermally degraded materials, e.g., polymethyl methacrylate (PMMA), polysulfone (PSF), high amylose starch (HAS), and phenolic resin (PR), can be used as template pore-forming agents [28,29,30,31]. Owing to the difference in thermal degradation behavior of pore-forming agents, the addition of these pore-forming agents has an important effect on the formation and physical and chemical structure of NFs and CNFs. The forming mechanism of PCNFs still needs further exploration.

In this work, we reported a facile template method to fabricate PCNFs using polyacrylonitrile (PAN) as a polymer carbon source and three different polymers PSF, HAS, and PR as pore-forming agents. PCNFs were prepared by electrospinning, pre-oxidation, and carbonization without additional treatment. The surface morphology, chemical components, graphitized crystallization, pore characteristics, and pore-forming mechanism were analyzed by scanning electron microscope (SEM), X-ray photoelectron spectroscopy (XPS), X-ray diffractometer (XRD), N_2_ physical adsorption, and thermal property, respectively. When PR was used as pore-forming agent, PCNF-R had a specific surface area as high as ~994 m^2^/g and a porosity as high as ~0.75 cm^3^/g. We also fabricated the supercapacitor electrodes using PCNFs as active materials and metal foam nickels as current collectors. The electrochemical performances, e.g., specific capacitance, rate capability, internal resistance, and cycling stability of PCNFs electrodes were studied using the electrochemical workstation under a three-electrode system. These hierarchical porous carbon nanofibers exhibit great potential in the development of polymer-derived low-cost and simple-process carbon materials for energy storage and other applications.

## 2. Results and Discussion

### 2.1. Surface Morphology

Owing to the different solubility in organic solvents, different fibrous morphology appears after electrospinning for polymer-blended precursors, i.e., PAN/PSF, PAN/HAS, and PAN/PR. PAN has good solubility in a variety of solvents, e.g., DMF or DMSO. However, HAS molecules with a large number of hydroxyl groups can easily form hydrogen bonds in the dissolution process, which results in poor solubility in DMF. By contrast, DMSO as a strong polar solvent can dissolve HAS to form a uniform solution blended with the other polymers. Therefore, the PAN/HAS precursor is fabricated using DMSO, as PAN/PSF and PAN/PF are fabricated using DMF as an organic solvent.

The SEM image and diameter distribution of electrospun nanofibers are shown in Figure 1. All nanofibers have smooth and irregular non-woven fiber morphology, and no beads are observed within the fibers. The fabricated fibers show nano-scale diameter. The diameter of NFs is 810 ± 280 nm. When the nanofibers contain 20 wt% PSF, HAS, and PR, the fibers become thinner. The diameters of NF-F, NF-S, and NF-R are 670 ± 180, 685 ± 335, and 560 ± 230 nm, respectively. Generally, the diameter of electrospun nanofibers is determined by polymer concentration, spinning parameters (e.g., voltage, distance, extrusion rate), and solution properties. Polymer concentration and spinning parameters remain unchanged during electrospinning. Therefore, the difference in fiber diameter can be attributed to the effect of the added polymer on the conductivity of the solution. When the pore-forming polymer is added to the PAN solution, the conductivity of the solution increases slightly. The increased conductivity of the solution can increase the charge density and the force of stretching the jet or filament, which leads to a decrease in the fiber diameter.

The SEM image and diameter distribution of electrospun carbon nanofibers are shown in Figure 2. Compared with nanofibers, carbon nanofibers have bending fiber shapes after heat treatment. The diameter of CNFs is 310 ± 130 nm, and the diameters of PCNF-F, PCNF-S, and PCNF-R become 230 ± 90, 250 ± 90, and 190 ± 45 nm, respectively. After adding the pore-forming polymer, the diameter of carbon nanofibers became smaller, and the diameter distribution became narrower. Furthermore, PCNF-R fibers have an obvious interconnected structure, which stems from the melting behavior of PR.

### 2.2. Chemical Components

Figure 3a shows the FTIR spectra of three pore-forming polymers (PSF, HAS, and PR) and nanofibers (NFs, NF-F, NF-S, and NF-R). The vibrational peak of NFs at 2245 cm^−1^ (-CN) and 1453 cm^−1^ (-CH_2_) represents the characteristic peak of PAN. For PSF, the two bands near 1580 and 1490 cm^−1^ are caused by the plane skeleton vibration of the aromatic ring. There are strong absorption peaks at 1260–1150 cm^−1^, representing the sulfonic group (S=O) in the PSF molecule. HAS has a C=O stretching vibration peak at 1730 and 1666 cm^−1^. For PR, the characteristic peaks are observed at 3300, 1500, and 1095 cm^−1^, respectively, representing the phenolic hydroxy functional group, the plane skeleton vibration of the benzene ring, and the C-H bending vibration in the plane of the benzene ring in the PR molecule. Nanofibers exhibit characteristic peaks containing PAN and pore-forming polymers, which indicate that pore-forming polymers are successfully added to nanofibers.

Figure 3b shows the full XPS spectra of carbon nanofibers, showing the element composition and content on the surface of carbon nanofibers. The fabricated carbon nanofibers are mainly composed of C, N, and O elements. As listed in Table 1, the C content is more than 90%, which indicates that the carbon nanofibers have a high carbonization degree. The content of N and O elements is low, in which N is provided by the nitrile group in the PAN molecule, and O is provided by the oxygen in the air during the pre-oxidation process. After adding pore-forming polymers, the C content increases significantly, and PCNF-F has the highest C content (95.6%), which indicates that the addition of pore-forming polymers is more conducive to carbonization. Compared with CNFs, the N and O content of PCNF-F, PCNF-S, and PCNF-R decrease, which indicates that the addition of pore-forming polymers has adverse effects on element doping.

### 2.3. Graphitized Crystallization

The graphitized structures of CNFs, PCNF-F, PCNF-S, and PCNF-R are analyzed by XRD. As shown in Figure 4, all XRD patterns reveal two main diffraction peaks of (100) and (002) at around 2*θ* = 16° and 2*θ* = 22°, respectively. The existence of the (100) diffraction peak indicates that the graphitization degree is insufficient. Compared with CNF, the (100) peak corresponding to PCNF-F, PCNF-S, and PCNF-R weakens, and the (002) peak strengthens, indicating that the graphitization degree has been improved. To further study the graphitized structure, the graphite crystalline thickness (*L*) and the inter-layer distance (*D*) at the (002) peak are calculated using the Scherrer and Bragg equations, and the results are listed in Table 1.
(1)$$L=\frac{k\lambda }{\beta \cos\theta }$$
(2)$$D=\frac{n\lambda }{2\sin\theta }$$

Here, *k* (0.89) is the Scherrer factor, *β* is the full width at half maximum at 2*θ* of approximately 25°, *λ* (0.15406 nm) is the wavelength of the X-rays, and *n* = 1.

PCNF-F has the largest *L* value of 1.31 nm, which indicates the highest graphitization degree among all the carbon samples. However, the D values of carbon nanofibers are very close, indicating that they have similar graphitization crystal forms. These results show that the addition of pore-forming polymers can induce the formation of a graphitized crystal structure in the high-temperature carbonization stage and improve the graphitization degree of carbon nanofibers. The graphitization degree often determines the conductivity of carbon materials. Furthermore, the resistivity of carbon nanofibers is measured by a resistivity tester, and the test results are listed in Table 1. With the addition of pore-forming polymers, the conductivity of carbon nanofibers has been significantly improved. PCNF-F has the highest conductivity at ~10.85 S/cm.

### 2.4. Pore Characteristics

As shown in Figure 5a–d, the BET specific surface area and pore structure are analyzed by N_2_ adsorption–desorption measurements. In contrast, PCNF-R shows the highest adsorption capacity, corresponding to the largest specific surface area. All the carbon nanofibers have a pore distribution range from 0.5 to 5 nm, which represents hierarchical pore structures containing micropores and mesopores. Table 2 shows the pore characteristics of carbon nanofibers. The specific surface areas of CNFs, PCNF-F, PCNF-S, and PCNF-R are 46, 687, 637, and 994 m^2^/g, respectively. The co-existence of micropores and mesopores in the carbon skeleton could supply a great number of active sites, reduce diffusion resistance, and facilitate the effective transfer of ions during charging and discharging [32].

### 2.5. Pore-Forming Mechanism

To explore the pore-forming mechanism, the thermal properties of nanofibers and pore-forming polymers are analyzed by DSC and TGA. As shown in Figure 6a, it can be seen from the DSC curve that there is an obvious exothermic peak at 285 °C in PAN, which represents the stabilization process of PAN. During this stage, PAN molecules form a stable trapezoidal ring structure. When the temperature rises further, a wider and weaker exothermic peak appears, representing the carbonization process of PAN. However, PSF, HAS and PR show obvious endothermic peaks, which are derived from the thermal decomposition behavior of pore-forming polymers. To further explore the effect of the thermal decomposition of pore-forming polymers on carbon nanofibers, TGA curves of PAN, PSF, HAS, and PR are tested, as shown in Figure 6b. The thermal decomposition of pure PAN nanofiber can be divided into three stages. The temperature of the first stage is about 100 °C, which is caused by the evaporation of water inside the PAN nanofiber. The second stage is in the range of 260~450 °C and mainly comes from the cyclization of the PAN nitrile group. The final stage is from 450 to 1000 °C, corresponding to the carbonization process. Compared with PAN, the pore-forming polymers have different thermal decomposition behaviors. PSF begins to decompose at 480 °C and rapidly decomposes at 480~600 °C, with a weight reduction of 65%. In contrast, HAS begins to decompose at 260 °C. When the temperature increases from 260 to 720 °C, the weight decreases by 75%. PR undergoes a continuous self-crosslinking reaction from 200 to 800 °C, including cyclization and carbonization, with a weight loss of more than 60%. Unlike PSF and HAS, PR exhibits a wide thermal decomposition behavior from low temperature to high temperature, which may be more conducive to the formation of pores in carbon nanofibers.

### 2.6. Electrochemical Performances

The electrochemical performances of CNFs, PCNF-F, PCNF-S, and PCNF-R electrodes are tested by three-electrode equipment. As shown in Figure 7a, all the electrodes have quasi-rectangular CV curves, indicating outstanding capacitive behaviors. Even at a scan rate range of 10–200 mV/s, the CV curves still maintain the “nearly rectangular shapes” (Figure 7b–e), suggesting low resistance and good reversibility. Moreover, the PCNF-R shows larger areas surrounded by CV curves than that of other electrodes, whether at low or high scan rates. These CV results indicate that PCNF-R has the highest capacitance value among all the electrodes.

The GCD curves of CNFs, PCNF-F, PCNF-S, and PCNF-R electrodes at the current density of 1 A/g are shown in Figure 8a. All the electrodes show symmetrical and nearly linear GCD curves, which suggests that they have good electrochemical reversibility and electric double-layer capacitance characteristics. In comparison, the PCNF-R electrode has a significantly longer discharge time than the other electrodes, which is consistent with the CV results. The specific capacitances of electrodes calculated by Equation (3) (see Experimental Section) at different current densities are shown in Figure 8b–e. The PCNF-R electrode has an evidently high specific capacitance of ~350 F/g. The specific capacitances of CNFs, PCNF-F, and PCNF-S are 96, 289, and 262 F/g at 1 A/g, respectively. As listed in Table 3, this capacitance value of PCNF-R is also higher than most of the other porous carbon nanofiber materials reported in the literature. The specific capacitance decreases with the increase in the current density (Figure 8f). When the current density is 20 A/g, the specific capacitance of the PCNF-R electrode remains as high as 254 F/g, with 72.6% retention at 1 A/g, indicating a good rate capability.

The electrical resistance and ion-transfer behavior of CNFs, PCNF-F, PCNF-S, and PCNF-R electrodes are tested by EIS (Figure 9a). The EIS fitting model and equivalent circuit are shown in Figure 9b, the semicircle intercept and the intercept at the Z’-axis represent the charge-transfer resistance (*R*_ct_) and the solution resistance (*R*_s_) in the high-frequency range. PCNF-R has a very small *R*_s_ value of ~0.55 Ω and the smallest semicircle diameter among all the electrodes, indicating low equivalent series resistance in the carbon nanofiber electrode. In the low-frequency range, the linear region (Warburg resistance) represents the ion-diffusion behavior of electrolytes within the electrodes. It is obvious that the Warburg curve of the PCNF-R electrode shifts toward the -Z’’-axis, which indicates that the PCNF-R has an excellent capacitive characteristic.

The cycling stability of all the electrodes is evaluated by the GCD cycling at a current density of 1 A/g. As shown in Figure 9c, after 10,000 cycles of charging and discharging, the retention rates of specific capacitances for CNFs, PCNF-F, PCNF-S, and PCNF-R electrodes are almost 100%. Such excellent cycling performance is due to the presence of microporous and mesoporous structures within carbon materials, which provide fast electronic transfer pathways [33].

The CV and GCD curves of the supercapacitor device made from PCNF-R electrodes were also measured. As shown in Figure 10a,b, the devices show similar trends to those obtained from the three electrode systems, except for the voltage ranges. The specific capacitance for the whole device is calculated based on the GCD curves. At a current density of 0.25 A/g, the specific capacitance of the PCNF-R-based device is 67.5 F/g, and it still maintains a high capacitance of ~43.5 F/g at a larger current density of 5 A/g (~64% of the capacity at 0.25 A/g) (Figure 10c).

## 3. Experimental Section

### 3.1. Materials

N,N-dimethylformamide (DMF) and dimethyl sulfoxide (DMSO) were purchased from Kemiou Chemical Reagent Co., Ltd, Tianjin, China. Polyacrylonitrile (PAN, molecular weight = 150,000), high amylose starch (HAS, molecular weight = 55,000), polysulfone (PSF, molecular weight = 50,000), phenolic resin (PR, molecular weight = 2800), polyvinylidene fluoride (PVDF) powder, and carbon black (CB) were purchased by Aladdin Chemical Reagent Co., Ltd, Shanghai, China.

### 3.2. Fabrication of PCNFs

Firstly, 3.2 g PAN and 0.8 g PSF (HAS, PR) were dissolved in 33 g DMF or DMSO. The blended solution was stirred for 6 h in oil bath at 60 °C. Secondly, the polymer precursor was electrospun into NFs. The voltage, distance, extrusion rate, and rotary speed were set as 20 kV, 15 cm, 1 mL/h, and 200 r/min, respectively. The obtained NFs were dried in oven at 60 °C for 2 h. Lastly, the dried NFs were pre-oxidized in a muffle furnace at 260 °C with a heating rate of 2 °C/min for 1 h under air atmosphere. The PCNFs were fabricated by carbonizing pre-oxidized NFs in a tubular furnace at 1000 °C with a heating rate of 5 °C/min for 2 h under nitrogen atmosphere. The fabricated NFs were marked as NF-F, NF-S, NF-R, and PCNFs were marked as PCNF-F, PCNF-S, PCNF-R (F = PSF, S = HAS, R = PR). Pure PAN nanofibers and carbon nanofibers were marked as NFs and CNFs.

### 3.3. Fabrication of Electrodes

Firstly, the PCNFs were further ground using an agate mortar, and a viscous slurry was fabricated by mixing PCNFs, PVDF powder, and CB at the mass ratio of 75:10:15 in DMF solvent. Secondly, the slurry was coated on the metal foam nickel and placed in the oven at 60 °C for 2 h to obtain the electrode. The metal foam nickel was washed using absolute alcohol at least three times and subsequently dried in the oven at 60 °C for 6 h. Lastly, fabricated electrodes were cut into squares, keeping the area of the electrode piece ~1 cm^2^ and the mass of the coated slurry ~1 mg.

### 3.4. Material Characterization

The surface morphology was observed using scanning electron microscope (SEM, Gemini SEM500, Carl Zeiss, Oberkochen, Germany). The chemical components were analyzed by X-ray photoelectron spectroscopy (XPS, Thermo Scientific K-alpha, Waltham, MA, USA). The graphitized crystallization was analyzed by an X-ray diffractometer (XRD, D8 ADVANCE, Bruker, Karlsruhe, Germany). The pore characteristics were analyzed using physical adsorption of N_2_ atmosphere at 77 K with an Autosorb-iQ-C apparatus (Quantachrome Instruments, Boynton Beach, FL, USA). Thermal gravimetric analysis (TGA, STA449F3, NETZSCH, Selb, Germany) and differential scanning calorimeter (DSC, 200F3, NETZSCH, Selb, Germany) were conducted to record the weight loss in an air atmosphere from room temperature to 1000 °C at a heating rate of 10 °C/min.

### 3.5. Electrochemical Measurements

The electrochemical performances of electrodes were measured by a CS2350H electrochemical workstation (Wuhan Corrtest, China). The PCNFs electrode, Pt foil, and Hg/HgO were used as the working electrode, the counter electrode, and the reference electrode, respectively. Then, 6 M KOH aqueous solution was used as electrolyte. The electrochemical impedance spectroscopy (EIS) was measured at the open circuit voltage with an amplitude of 5 mV in the frequency range of 0.01 to 10 kHz. Cyclic voltammetry (CV) at the scan rate range of 10–200 mV/s and galvanostatic charge–discharge (GCD) at the current density range of 1.0–50 A/g were conducted with the potential range of −1–0 V. The specific capacitances of the electrodes were calculated from GCD data based on the following equation. The cycle stability of electrodes was evaluated at 1 A/g for 10,000 cycles.
(3)$$C_{m}=\frac{I\times \Delta t}{m\times \Delta V}$$
where *C*_m_ is the mass specific capacitance (F/g), *I* is the response current (A), Δ*t* is the discharging time (s), *m* is the mass of activated material (mg), and *ΔV* is the potential window (V).

The supercapacitor device was fabricated using two symmetric electrodes and a piece of PP/PE composite separator. The CV of the device was measured between 0 and 1.0 V for a 6 M KOH aqueous electrolyte by varying the scan rate from 10 to 200 mV/s. The GCD of the device at a current density of 0.25–5 A/g was tested using the CS2350H electrochemical workstation, and the GCD specific capacitances were calculated using the following equation.
(4)$$C_{d}=\frac{I\times ∆t}{M\times ∆V}$$
where *C*_d_ (F/g) is the specific capacitance of device, Δ*V* (V) is the potential change within the discharge time Δ*t* (s), and *M* is the total mass of active materials in the two electrodes.

## 4. Conclusions

The template method is one of the important methods used to fabricate PCNFs. However, there are few reports about the effects of different pore-forming polymers on the structure and properties of PCNFs. In summary, we have demonstrated PCNFs fabricated via a facile and low-cost method and employed as electrode-active materials for supercapacitor applications. Three different pore-forming agents, i.e., PSF, HAS, and PR, were added to the nanofibers. The results show that the addition of pore-forming agents is beneficial to the improvement of graphitization degree. The thermal degradation behavior of pore-forming agents has a great influence on the formation of pores. Among them, PR is more suitable as a pore-forming agent due to its wide thermal decomposition behavior. PCNF-R has a specific surface area as high as ~994 m^2^/g, a total pore volume as high as ~0.75 cm^3^/g, and a good graphitization degree. When PCNF-R are used as active materials to fabricate into electrodes, the PCNF-R electrode shows excellent electrochemical performances, a high specific capacitance ~350 F/g, a good rate capability ~72.6%, a low internal resistance ~0.55 Ω, and an excellent cycling stability ~100% after 10,000 charging and discharging cycles.

## Figures and Tables

**Figure 1 molecules-28-02228-f001:**
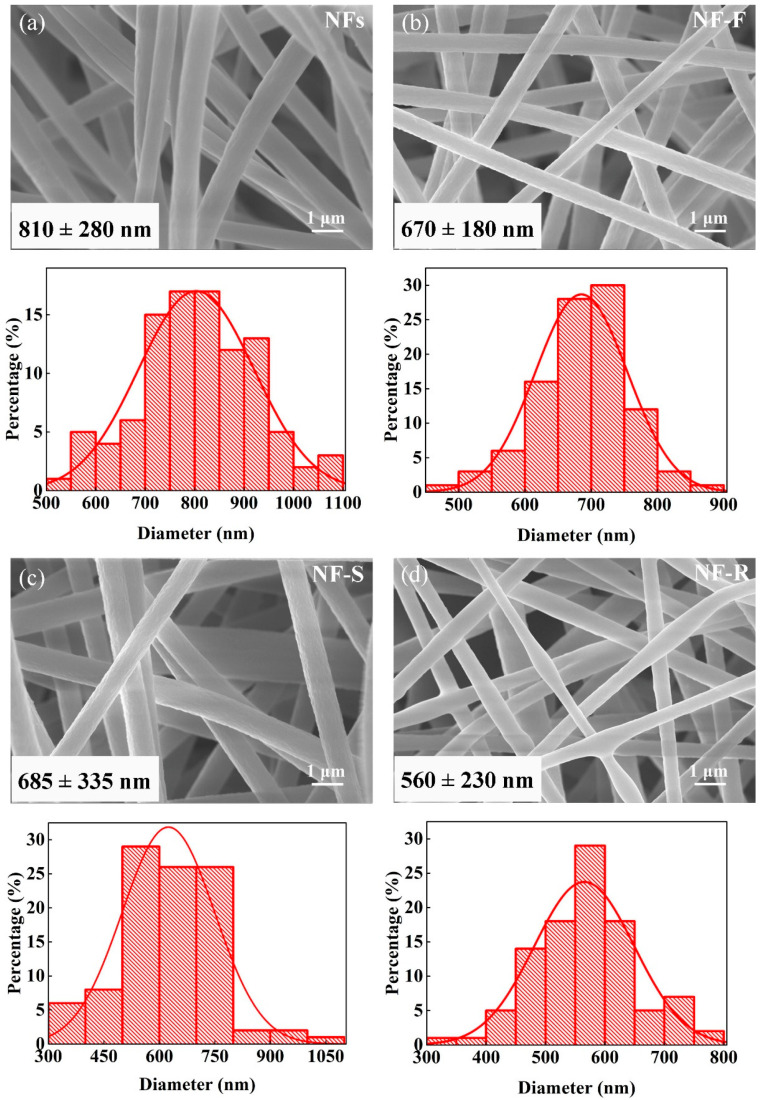
SEM images and diameter distribution of (**a**) NFs, (**b**) NF-F, (**c**) NF-S, and (**d**) NF-R.

**Figure 2 molecules-28-02228-f002:**
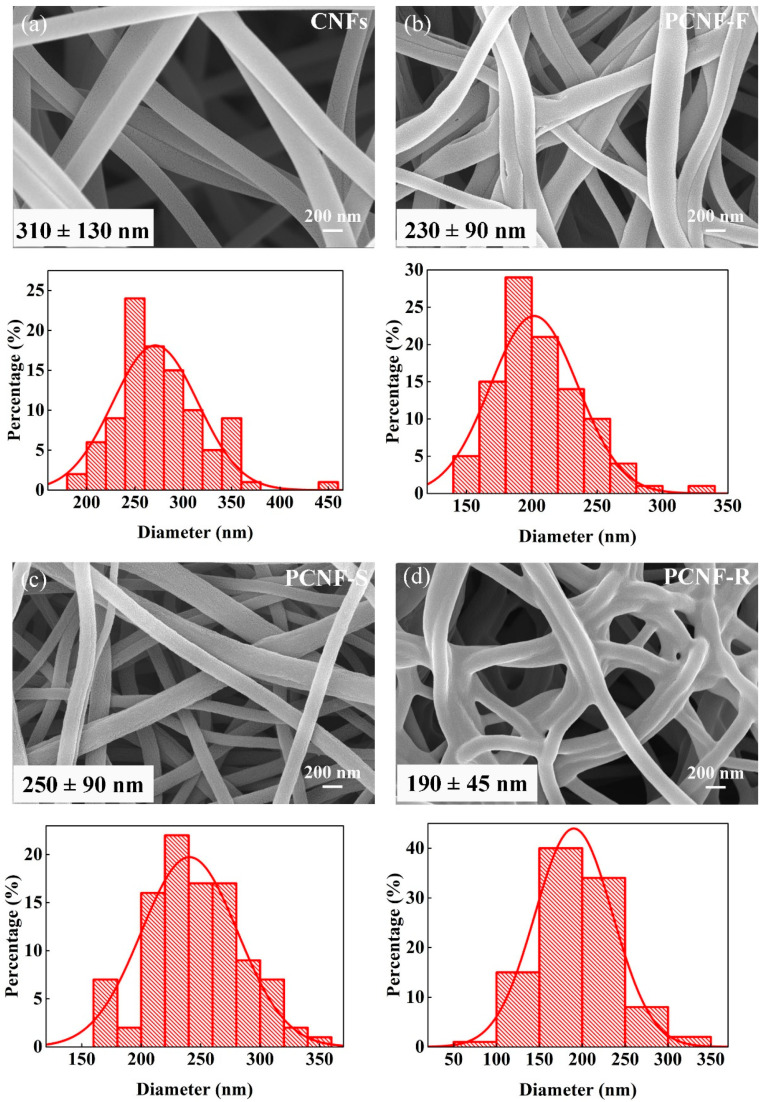
SEM images and diameter distribution of (**a**) CNFs, (**b**) PCNF-F, (**c**) PCNF-S, and (**d**) PCNF-R.

**Figure 3 molecules-28-02228-f003:**
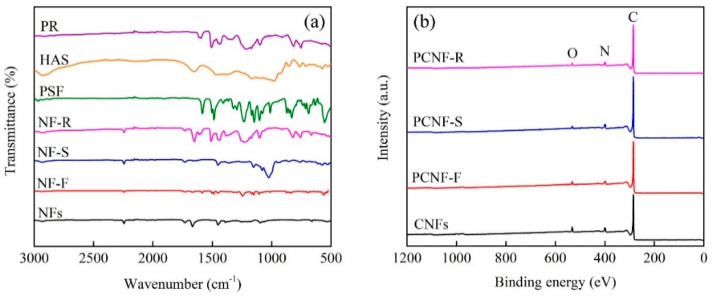
(**a**) FTIR spectra of PSF, HAS, PR, NFs, NF-F, NF-S, and NF-R. (**b**) XPS spectra of CNFs, PCNF-F, PCNF-S, and PCNF-R.

**Figure 4 molecules-28-02228-f004:**
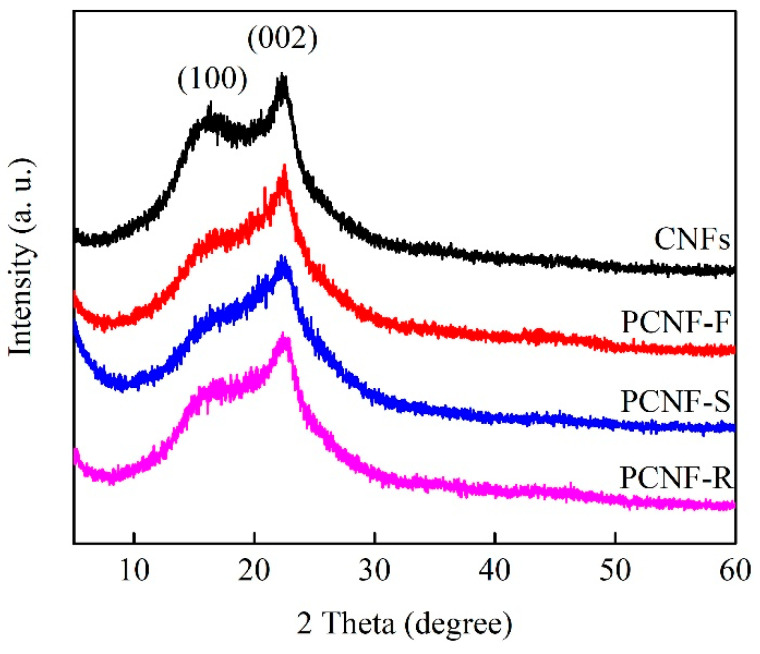
XRD spectra of CNFs, PCNF-F, PCNF-S, and PCNF-R.

**Figure 5 molecules-28-02228-f005:**
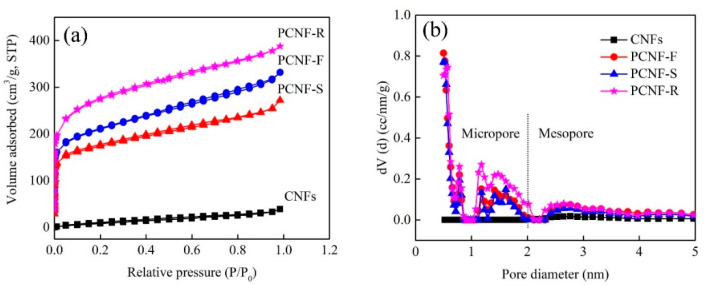
(**a**) N_2_ adsorption–desorption isotherm of CNFs, PCNF-F, PCNF-S, and PCNF-R. (**b**) Pore-size distribution of CNFs, PCNF-F, PCNF-S, and PCNF-R.

**Figure 6 molecules-28-02228-f006:**
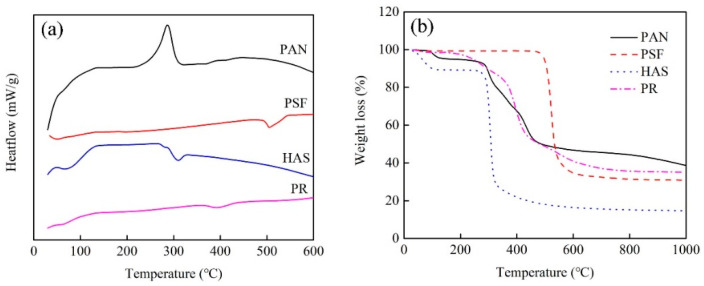
(**a**) DSC curves of PAN, PSF, HAS, and PR. (**b**) TGA curves of PAN, PSF, HAS, and PR.

**Figure 7 molecules-28-02228-f007:**
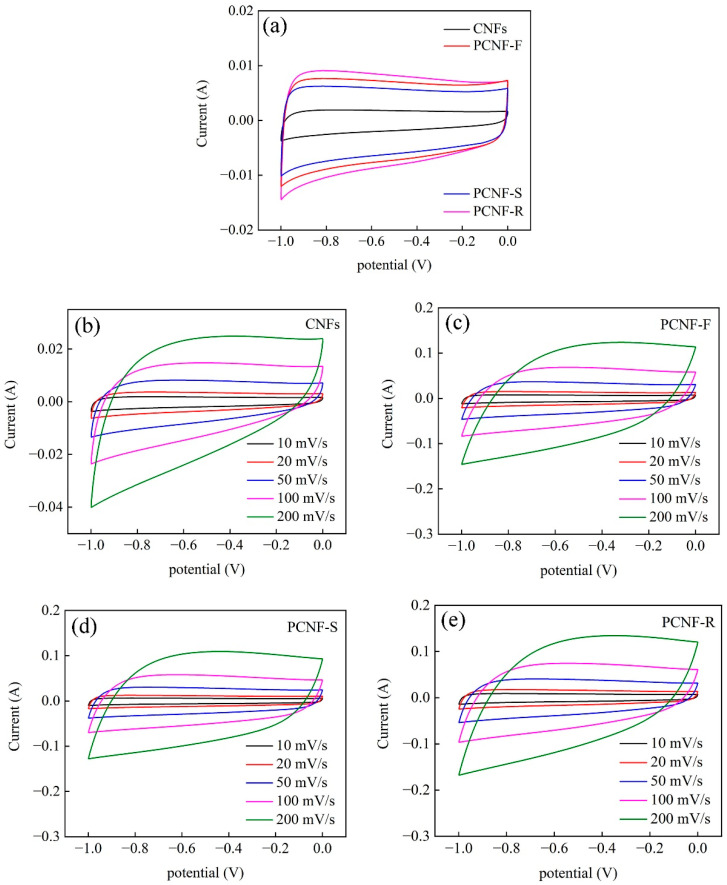
(**a**) CV curves of CNFs, PCNF-F, PCNF-S, and PCNF-R electrodes under the scan rate of 10 mV/s. CV curves of (**b**) CNFs, (**c**) PCNF-F, (**d**) PCNF-S, and (**e**) PCNF-R at the scan rate range of 10–200 mV/s.

**Figure 8 molecules-28-02228-f008:**
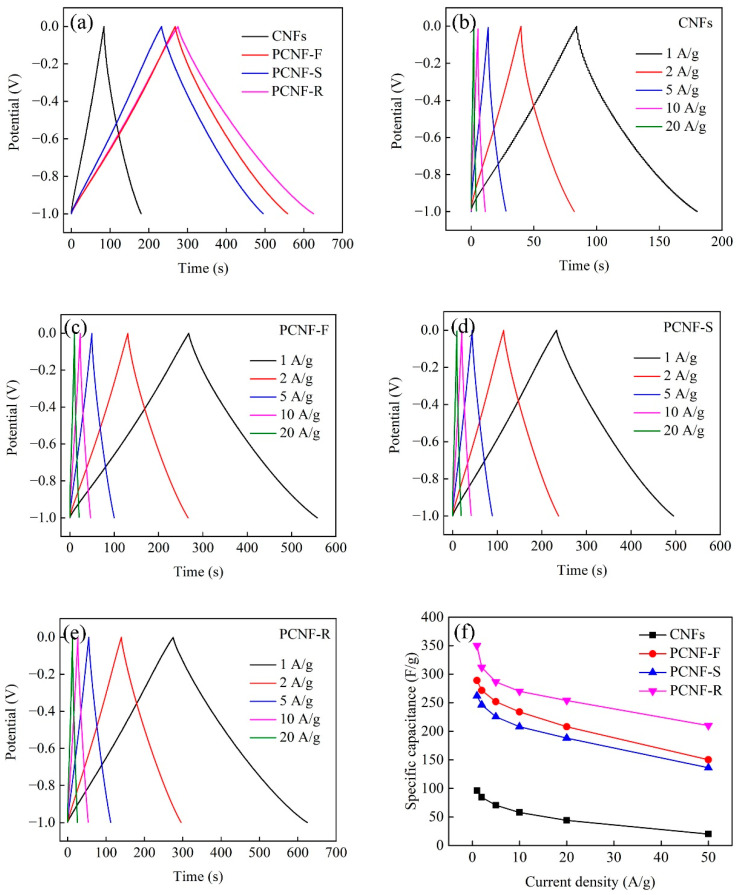
(**a**) GCD curves of CNFs, PCNF-F, PCNF-S, and PCNF-R electrodes under the current density of 1 A/g. GCD curves of (**b**) CNFs, (**c**) PCNF-F, (**d**) PCNF-S, and (**e**) PCNF-R at the current density range of 1–20 A/g. (**f**) Specific capacitances of CNFs, PCNF-F, PCNF-S, and PCNF-R electrodes at different current densities.

**Figure 9 molecules-28-02228-f009:**
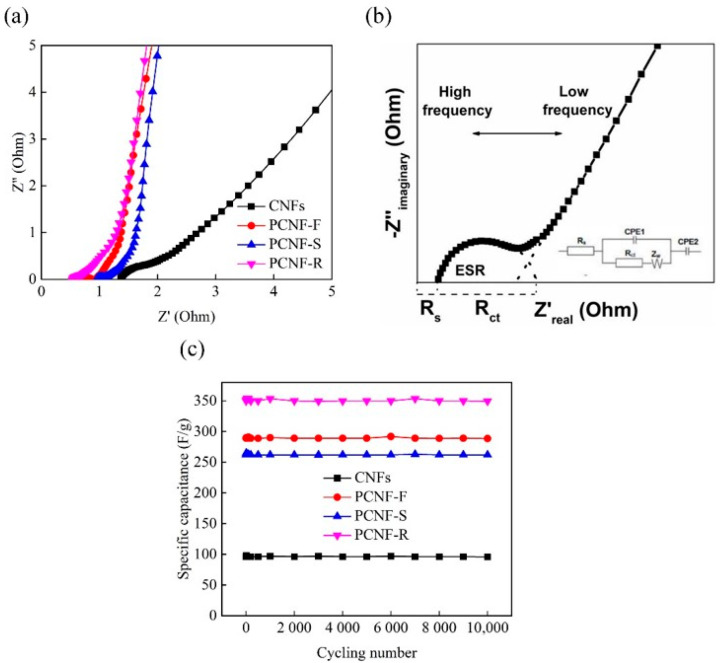
(**a**) Nyquist plots of CNFs, PCNF-F, PCNF-S, and PCNF-R electrodes. (**b**) The EIS fitting model and equivalent circuit of the electrode. (**c**) Cycling stability of CNFs, PCNF-F, PCNF-S, and PCNF-R electrodes at a current density of 1 A/g.

**Figure 10 molecules-28-02228-f010:**
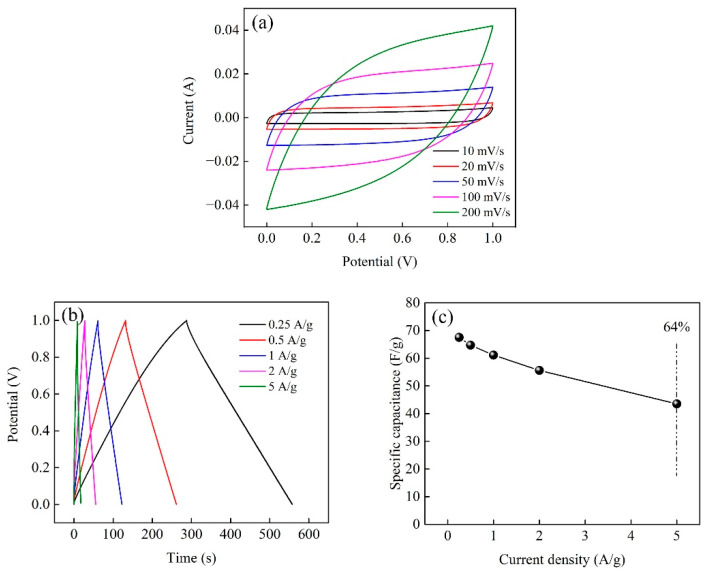
(**a**) CV curves of device made from PCNF-R electrodes under the scan rate of 10 mV/s. (**b**) GCD curves of device made from PCNF-R electrodes under the current density of 0.25 A/g. (**c**) Specific capacitances of device made from PCNF-R electrodes at different current densities.

**Table 1 molecules-28-02228-t001:** XPS, XRD and conductivity results of carbon nanofibers.

Samples	C	N	O	*D* (Å)	*L*	Conductivity
	(%)	(%)	(%)		(nm)	(S/cm)
CNFs	90.0	5.1	4.9	4.01	0.87	6.64
PCNF-F	95.6	2.7	1.7	3.95	1.31	10.85
PCNF-S	94.5	3.5	2.0	3.99	1.12	9.32
PCNF-R	94.2	4.1	1.7	3.97	1.06	7.84

**Table 2 molecules-28-02228-t002:** Pore characteristics results of carbon nanofibers.

Samples	Specific Surface Area	Porosity	Mesopore Volume	Mesopore Content	Micropore Volume	Micropore Content
	(m^2^/g)	(cm^3^/g)	(cm^3^/g)	(%)	(cm^3^/g)	(%)
CNFs	46	0.06	0.05	83	0.01	17
PCNF-F	687	0.52	0.22	42	0.30	58
PCNF-S	637	0.42	0.17	41	0.25	59
PCNF-R	994	0.75	0.20	27	0.55	73

**Table 3 molecules-28-02228-t003:** Specific capacitance of electrospun carbon nanofibers reported the literature.

Precursor	Pore-Forming Agent	Precursor to Activation Agent (Mass Ration)	Optimum Capacitance	Ref.
			(F/g)	
PAN	PSF	4:1	289 (1 A/g)	This work
PAN	HAS	4:1	262 (1 A/g)	This work
PAN	PR	4:1	350 (1 A/g)	This work
PAN	PMMA	4:1	260 (1 A/g)	[34]
PAN	PVP	4:1	225 (1 A/g)	[34]
PAN	Nafion	1:4	210 (1 A/g)	[20]
Lignin	Mg(NO_3_)_2_	1:2	248 (0.2 A/g)	[35]
PAN	CaCO_3_	3:8	251 (0.5 A/g)	[22]
PVA	PTFE	1:15	177 (1 A/g)	[36]
PAN	ZnCl_2_	9:5	214 (1 A/g)	[37]
Cellulose	CO_2_	-	241.4 (1 A/g)	[38]
PAN	PMMA	1:9	243 (1 A/g)	[39]

## Data Availability

The data are contained within the article.

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
