# Peer review of "Fabrication of Porous Carbon Nanofibers from Polymer Blends Using Template Method for Electrode-Active Materials in Supercapacitor"

_molecules, 2023, doi:10.3390/molecules28052228_

Round 1

Reviewer 1 Report

The focus of this article is the Fabrication of porous carbon nanofibers from polymer blends using template method for electrode active materials in supercapacitor. Further, fabricate into electrodes, the PCNF-R electrode show a high specific capacitance 350 F/g, a good rate capability 72.6%, a low internal resistance 0.55 Ω, and an excellent cycling stability 100% after 10000 charging and discharging cycles. The manuscript is well written with enough characterization results. However, manuscript requires improvement. I do recommend this manuscript reconsider after a revision. Detail comments are included below.

1.      What is the actual mass of the electrode materials?

2.      Why the authors do not carry out the 2-elecrode system? If possible, add the device results either symmetric or asymmetric.

3.      Is there any pre-treatment was done for nickel foam prior to the electrode fabrications?

4.      In Fig. 9(a), EIS data should be fit and include the equivalent circuit diagram.

5.      Authors should cite the following recent papers in the introduction section of the revised manuscript.

Molecules 2022, 27(22), 7691; Nanomaterials 2022, 12(9), 1511.

Author Response

Reviewer 1

The focus of this article is the Fabrication of porous carbon nanofibers from polymer blends using template method for electrode active materials in supercapacitor. Further, fabricate into electrodes, the PCNF-R electrode show a high specific capacitance 350 F/g, a good rate capability 72.6%, a low internal resistance 0.55 Ω, and an excellent cycling stability 100% after 10000 charging and discharging cycles. The manuscript is well written with enough characterization results. However, manuscript requires improvement. I do recommend this manuscript reconsider after a revision. Detail comments are included below.

  1. What is the actual mass of the electrode materials?

Response: revised.

The actual mass of the electrode material is ~ 1 mg. Please see the Experimental Section, “3.3. Fabrication of electrodes”.

  1. Why the authors do not carry out the 2-elecrode system? If possible, add the device results either symmetric or asymmetric.

Response: revised.

We have supplemented the 2-electrode measurement results in the symmetric device. Please the CV and GCD curves in Figure 10.

  1. Is there any pre-treatment was done for nickel foam prior to the electrode fabrications?

Response: revised.

We have supplemented relative contents in the Experimental section, “3.3. Fabrication of electrodes”. “The metal foam nickel was washed using absolute alcohol at least three times and subsequently dried in the oven at 60 ℃ for 6 h.”

  1. In Fig. 9(a), EIS data should be fit and include the equivalent circuit diagram.

Response: revised.

We have supplemented relative contents. Please see the Fig. 9(b).

  1. Authors should cite the following recent papers in the introduction section of the revised manuscript. Molecules 2022, 27(22), 7691; Nanomaterials 2022, 12(9), 1511.

Response: revised.

We have cited these papers in the References section.

Reviewer 2 Report

In the study titled "Fabrication of porous carbon nanofibers from polymer blends using template method for electrode active materials in supercapacitor", it is seen that there is a study to obtain effective Porous carbon nanofibers (PCNFs) structures. Looking at the literature, it is easy to see that there are many studies on "Porous carbon nanofibers". The synthesis method of the present study seems suitable for this study. However, considering the evaporation temperature of the DMF solvent, which is used as a solvent in the production of electrodes, the preferred temperature and time seemed a little thought-provoking to me. Why has not the carbonization method been done in an inert environment? Previous studies at https://doi.org/10.1007/s10854-022-09264-9 and https://doi.org/10.1016/j.colsurfa.2021.127787 can be reviewed. Lastly, the amount of active material used in electrochemical measurements is about 1-2 mg. What amount of substance did the author use in equation 3?  According to Equation 3, if the amount of substance is chosen as 2 mg instead of 1 mg, the amount of capacitance to be obtained decreases by half. Therefore, the exact amount of substance used should be given. 

Author Response

Reviewer 2

In the study titled "Fabrication of porous carbon nanofibers from polymer blends using template method for electrode active materials in supercapacitor", it is seen that there is a study to obtain effective Porous carbon nanofibers (PCNFs) structures. Looking at the literature, it is easy to see that there are many studies on "Porous carbon nanofibers". The synthesis method of the present study seems suitable for this study.

  1. However, considering the evaporation temperature of the DMF solvent, which is used as a solvent in the production of electrodes, the preferred temperature and time seemed a little thought-provoking to me.

Response: The DMF solvent is a very good material to prepare electrodes in our study. As we all know, DMF always volatilizes fast at room temperature. After we make the slurry using DMF solvent, we need to coat it onto the metal foam nickel immediately. At present, we have not conducted research on the influence of time and temperature on DMF-based slurry. However, we will consider it in the later work. Thank you again for your suggestions.

  1. Why has not the carbonization method been done in an inert environment?

Response: revised.

Please see the Experimental Section, “Please see the Experimental Section, “3.3. Fabrication of electrodes”. “Lastly, the dried NFs were pre-oxidized in a muffle furnace at 260 ℃ with a heating rate of 2 ℃/min for 1 h under air atmosphere. The PCNFs were fabricated by carbonizing pre-oxidized NFs in a tubular furnace at 1000 ℃ with a heating rate of 5 ℃/min for 2 h under nitrogen atmosphere.”

  1. Previous studies at https://doi.org/10.1007/s10854-022-09264-9 and https://doi.org/10.1016/j.colsurfa.2021.127787 can be reviewed.

Response: revised.

We have cited these papers in the References section.

  1. Lastly, the amount of active material used in electrochemical measurements is about 1-2 mg. What amount of substance did the author use in equation 3? According to Equation 3, if the amount of substance is chosen as 2 mg instead of 1 mg, the amount of capacitance to be obtained decreases by half. Therefore, the exact amount of substance used should be given.

Response: revised.

The actual mass of the electrode material is ~ 1 mg. Please see the Experimental Section, “3.3. Fabrication of electrodes”.

Reviewer 3 Report

The manuscript "Fabrication of porous carbon nanofibers from polymer blends using template method for electrode active materials in supercapacitor" is well written addressing every necessary detail.

The article may be accepted after properly addressing the following points:

1) Recent articles related to carbon nanofiber-based supercapacitors have to be discussed in the introduction part.

2) A comparison of the results obtained from this work with those in previous literature will give an insight into the performance of the fabricated electrodes.

Author Response

Reviewer 3

The manuscript "Fabrication of porous carbon nanofibers from polymer blends using template method for electrode active materials in supercapacitor" is well written addressing every necessary detail.

The article may be accepted after properly addressing the following points:

  1. Recent articles related to carbon nanofiber-based supercapacitors have to be discussed in the introduction part.

Response: revised.

Please see the Introduction section. “In the early research, although high specific surface areas can be obtained by physical method (CO2, steam as activator), the specific capacitance of the electrode is lower than 300 F/g due to the tedious microporous structure. To enhance the specific capacitance of PCNFs-based electrode, the chemical method is developed. PCNFs prepared by chemical activators such as KOH, H3PO4, and ZnCl not only have high specific surface areas and rich micro-meso pore structures but also have active elements doped in carbon nanofibers, effectively improving specific capacitance. However, the excessive use of chemical activator is not conducive to environmental protection, and the strong corrosivity will damage the equipment. These adverse factors limit the development of chemical activators.”

  1. A comparison of the results obtained from this work with those in previous literature will give an insight into the performance of the fabricated electrodes.

Response: revised.

We have supplemented relative contents. Please see the Table 3.